# Probiotics for the Prophylaxis of Migraine: A Systematic Review of Randomized Placebo Controlled Trials

**DOI:** 10.3390/jcm8091441

**Published:** 2019-09-11

**Authors:** Malwina M. Naghibi, Richard Day, Samantha Stone, Ashton Harper

**Affiliations:** ADM Protexin, Lopen Head TA13 5JH, UK; samantha.stone@adm.com (S.S.); ashton.harper@adm.com (A.H.)

**Keywords:** migraine, probiotics, prophylaxis, gut–brain axis, systematic review

## Abstract

Migraine is a common and disabling neurological condition with a complex etiology. Recent advances in the understanding of the gut microbiome have shown the role of gut micro-organisms in disease outcomes for distant organs—including the brain. Interventions targeting the gut microbiome have been shown to be effective in multiple neurological diagnoses, but there is little research into the role of the microbiome in migraine. This systematic review seeks to assess the current research landscape of randomized placebo controlled trials utilizing probiotic interventions as migraine prophylaxis. Searches were conducted of scientific databases including PubMed, MEDLINE, and the Cochrane Library, following PRISMA guidelines. Of 68 screened studies, 2 were eligible for analysis. Due to methodological differences, meta-analysis was not possible. Qualitative comparison of the studies demonstrated a dichotomy of results—one trial reported no significant change in migraine frequency and intensity, while the second trial reported highly significant improvements. No clear ‘gold standard’ currently exists for microbiome research, let alone for migraine-related microbiome research. The heterogeneity of outcome measures used in the two trials included in this systematic review shows the need for a standardization of outcome measures, therefore a series of recommendations for future probiotic–migraine research are included.

## 1. Introduction

Migraine headache is the third most common disease in the world, affecting approximately 15% of the global population [1]. Migraines are characterized by recurring severe unilateral throbbing headaches that last from four hours to three days and typically involve sensory disturbance to light (photophobia) and sound (phonophobia). Migraine headaches are categorized in different ways based on (1) presence of aura symptoms (visual disturbances or numbness; ~30% of migraineurs) or absence of aura symptoms (~70%) and (2) monthly burden: episodic migraine (<15 headache days per month; >90%) and chronic migraine (15 or more headache days per month; <10%) [2]. Migraines are substantially more prevalent in women (3:1) than men [3,4], which is strongly suggested to be due to fluctuating levels of female sex hormones during the reproductive years. The pathophysiology of migraine is complex and involves genetic, metabolic, and hormonal elements which disrupt the ability of the brain to process incoming sensory information [5].

The gastrointestinal microbiota—the vast collection of microorganisms residing in the gut—is implicated in the function and dysfunction of multiple extra-intestinal organs including the brain, and indeed have a recently suggested role in migraine headaches [6].

The microbiome–gut–brain axis consists of at least three bidirectional biochemical signaling pathways between the gut and the brain: nervous, endocrine, and immune [7].

Mechanisms as to how the microbiota and gut–brain axis interact are only beginning to be elucidated but may include alterations in microbial composition, immune activation, vagus nerve signaling, and production of specific microbial neuroactive metabolites [8].

The first direct evidence of behavioral effects induced by specific bacterial strains came from the animal murine model [9]. When a strain of *Bifidobacterium longum* was orally administered to mice with established anxiety it normalized their behavior; provided their vagus nerves were intact. These findings demonstrated that the vagus nerve acts as a conduit for the anxiolytic effects of *B. longum*.

Serotonin (5-HT) and the serotonergic system are integral to the pathophysiology of migraine, as evidenced by the preponderance of serotonergic drugs used to treat and prevent migraines (e.g., triptans and tricyclic antidepressants). While it has been proposed that there are low levels of 5-HT before and between migraine attacks, with high levels being seen during attacks, the evidence for this is inconclusive between studies [10,11,12]. Additionally, serotonin functions as a signaling molecule in many gastrointestinal (GI) functions, including peristalsis, secretion, vasodilation, and perception of pain and nausea, the latter being symptoms often experienced by migraineurs [6,13]. This is interesting considering the concomitance between migraines and gastrointestinal (GI) disorders such that people who frequently experience gastrointestinal disorders have a higher prevalence of headaches [6,14].

Another gut–brain axis link involves increased gut permeability leading to inflammation and a release of pro-inflammatory cytokines [15]. Cytokines are important mediators of the immune and inflammatory pathways linking distant organs such as the GI and central nervous systems [16]. Elevated levels of chemokines may result in the stimulation and activation of trigeminal nerves (a key structure in migraine), and the release of vasoactive peptides and other biochemical mediators, such as nitric oxide, which ultimately lead to inflammation [16] and the generation of migraines.

Stress has been identified as the most common trigger for migraine attacks [17], and evidence suggests there may be an alteration in the hypothalamic–pituitary–adrenal axis in these patients. The stress response involves the release of a number of hormones including cortisol, which is known to increase the permeability of the gut barrier allowing the translocation of bacteria and pro-inflammatory compounds into the blood [18]. One of these compounds—lipopolysaccharide—is a cell wall component of gram negative bacteria with a well-documented pro-inflammatory effect. The relevance of lipopolysaccharides (LPS) to migraine headache is highlighted by evidence that it sensitizes pain receptors in trigeminal sensory neurons (those involved in migraine headaches) [19]. Research has identified that probiotics are capable of resolving stress-induced gut leakiness in rodents [20], and reducing cortisol levels in humans [21], evidence which may explain an additional mechanism by which they can influence migraine headaches.

The primary goals of migraine treatments include relieving pain, restoring function, and reducing headache frequency with current treatments for migraine centering on decreasing the frequency, severity, and duration of migraine attacks [22,23,24]. Recently, probiotics have become a point of focus for treating migraines considering the multiple mechanisms by which GI microorganisms may affect the function of the central nervous system (CNS). Work in this area has mainly focused on murine models or uncontrolled small sample sized studies, however there have been emerging studies published consisting of larger placebo controlled randomized trials [25]. The present systematic review aims to discuss the effectiveness of probiotic interventions in treating migraine symptoms based on published randomized controlled trials.

## 2. Materials and Methods

We performed a systematic review according to the PRISMA guidelines. We systematically searched the literature using the search string “probiotics AND migraine” using a range of search words for both terms (probiotics, Lactobacillus, Bifidobact*, Bacillus*, headache, migraine, cephalgia, cephalalgia). The list of studies was obtained by searching clinical databases including EMBASE, MEDLINE, PubMed, and the Cochrane library, up to January 2019. The clinical trial register, ClinicalTrials.gov, was searched for any ongoing or recently finished studies. Only publications in English were included and citation lists of relevant papers were checked. The systematic review was registered with PROSPERO (registration number CRD42018109458).

Retrieved studies were independently screened for relevance (title and abstract) by two reviewers and potentially relevant articles were retrieved. Studies were considered eligible if they met the following criteria: randomized controlled trials of adult male and female participants diagnosed with migraine based on defined clinical criteria, who were administered any probiotic formulation (and only probiotic) for a minimum of four weeks. The intervention had to be compared against a control group or placebo with results reported for frequency of migraine attacks and/or severity of migraine attacks at least at baseline and at the end of the intervention.

Two reviewers independently selected trials to be included; any disagreements were resolved by consensus. Each included study was assessed using the Jadad scale [26] in order to evaluate reporting on the method of randomization, adequacy of allocation concealment, blinding, and proportion and reporting of those lost to follow-up. One reviewer extracted the data from all formally included trials, and data cleaning of the generated database was performed by the second reviewer.

Data for the following symptoms were extracted from the included articles: demographics, migraine diagnostic criteria, migraine symptoms (severity, frequency, duration), quality of life or disability scores, use of medications, inflammation markers, dropout rates, and adverse events, if available. Considering that different bacterial strains perform different functions, details of the probiotic intervention was extracted, including strains, doses, and duration of the intervention.

## 3. Results

Ninety-seven titles were retrieved from all the searches and after removing duplicates, there were 67 original papers. Studies were excluded during screening (*n* = 57) for various reasons: different conditions, pediatric cohorts, different interventions, healthy population, or other (animal trial, non-English, review article), as shown in Figure 1 (PRISMA flow diagram). During the full text review of the remaining 10 papers, reasons for exclusion included: non-randomized controlled trial (non-RCT) design (4/8), abstract only (2/8), an open label study design (1/8), and different interventions (1/8).

Description of included studies can be found in Table 1, while risk of bias analysis and description of excluded studies can be found in Appendix A, respectively. Within the two included studies, a total of 163 patients were enrolled and 139 (113 women) completed trials; with 74 patients in probiotic groups and 65 in placebo. There was a substantial difference in the proportion of male and female participants in the trials with a 14:1 ratio in de Roos’ cohort and 2.7:1 in Martami’s cohort. In both studies, patients were screened by expert neurologists using adequate diagnostic criteria. Studies applied similar exclusion criteria—use of antibiotics or probiotics prior to enrolment (different time frames between the trial were used) and pregnancy or breastfeeding. In addition, de Roos excluded patients with chronic migraine or with inflammatory bowel disease, while Martami excluded patients using antipsychotic medications prior to enrolment.

The two trials used different multistrain probiotic formulations with similar doses (5 × 10^9^ vs. 4 × 10^9^ colony forming units), and the intervention details can be found in Table 1. Interventions were 12 or 8–10 weeks long with once-daily supplementation of probiotics or placebo. Only de Roos reported compliance, which was 94%.

The identified trials were not suitable for meta-analysis (due to small numbers and different reporting styles), therefore only descriptive analysis has been performed. Both authors were contacted with requests for their original data sets—to better understand if a meta-analysis of the data might be possible. Only one research group provided further data, and this did not facilitate a meta-analysis.

The studies reported over 30 clinical parameters each, with only 5 outcome parameters reported in the same or similar manner. The only results directly comparable between the two studies were inflammatory markers (TNFα, CRP) at baseline and at the end of the intervention. Neither study found any substantial change in inflammatory markers during the intervention, as shown in Table 2, with the exception of a small rise in CRP seen in the probiotic arm of the chronic migraine (CM) group in the Martami study.

Both studies used daily diaries to obtain information about migraine frequency and intensity; neither of which were validated. Migraine intensity was reported in a similar manner, with both trials using a 10-point visual analogue scale; however, de Roos reported an average from the daily pain scores higher than 4 points, while Martami reported the average of the total visual analogue scale scores. For that reason, values were not entirely comparable and suggest that Martami’s cohort had more severe migraine attacks, as the total average at baseline was higher in both chronic (8.5 in probiotic and 8.9 in placebo) and episodic groups (7.3 in probiotic and 6.4 in placebo), when compared with de Roos’ episodic baseline score (6.6 in probiotic and 6.4 in placebo).

De Roos demonstrated no change in migraine symptoms after the intervention, while the other study reported significant improvement in the migraine symptoms in both chronic and episodic migraine sufferers.

Frequency of migraine attacks (migraine days) at baseline were reported based on a one month (Martami) or three months (de Roos) analysis, suggesting that the de Roos cohort had a similar frequency of migraine attacks when compared to the episodic group in Martami’s paper. However, reporting of the effect of the intervention on migraine days was different between the studies. De Roos’ reporting included results for each 4 week period, with results ranging from 1 to 18 days in the probiotic arm and 0 to 15 migraine days in the placebo group. This was consistent with the results reported for the migraine disability assessment test (MIDAS) questionnaire by the same author (total migraine days over a 3 month period) suggesting 15 migraine days over 3 months both at baseline and at the end of the trial. Martami reported the effect of the intervention on the migraine days at the end of the trial. Intervention in the de Roos trial did not translate into a significant reduction in migraine days at any point of the trial, while Martami demonstrated a reduction of 9.6 migraine days in the CM group and 2.6 days reduction in the episodic migraine (EM) group, which was significantly different to results seen in the placebo arms.

Migraine days per month reduced in EM and CM groups receiving probiotics, which was significantly different to baseline and to placebo as reported by Martami, but de Roos showed no difference in the median number of migraine days between probiotic and placebo arms.

The change in severity of migraine attacks was also different between the studies. Martami found a reduction in the severity of migraine episodes in EM (mean change −2.1 points) and in CM (mean change −2.7 points), which was significantly different to respective placebo groups. Migraine intensity has remained fairly stable through the study and did not differ between probiotic and placebo in de Roos’ study.

De Roos did not report on migraine duration (despite the fact that it was a question included in the daily diary), whereas Martami reported that migraine duration remained unchanged in the EM cohort, while CM patients had a significant decrease in duration of migraine attacks compared to placebo (−0.59 hours).

De Roos included additional analysis of responders and non-responders, showing that more patients in the probiotic arm had a reduction of 2 or more days in migraine frequency from baseline to the end of the trial; 39% of patients (12 out of 31) in the probiotic arm and 24% (7 out of 29) of patients in the placebo group.

Use of medications was reported by both authors; however, de Roos reported it as a dosage of medication groups (triptans, paracetamol, combined analgesics, non-steroidal anti-inflammatory drugs), while Martami reported number of abortive drugs used before and after the intervention, with the percentage of patients using different medication groups at baseline (triptans, non-steroidal anti-inflammatory drugs, other analgesics, propranolol, topiramate, sodium valproate, tricyclic antidepressants). This inconsistency of reporting prevented direct comparison and understanding if cohorts had similar use of medications or a similar change in the use as a result of interventions. De Roos reported that use of medications had not differed between the groups or over time, while Martami reported a significant reduction in the use of abortive medications in EM (−0.72 dose/week) and in CM groups (−1.02 dose/day) taking probiotics vs. placebo. In addition, de Roos reported the number of patients using prophylaxis (13 patients) and the number of patients who increased (2 patients) or reduced (3 patients) their use during the study.

Only one paper reported on the effect of the intervention on quality of life. Baseline disability in the de Roos study was described as ‘severely limiting’, which improved to ‘moderately limiting’ at the end of the trial, however, time and group effects were not significant. Based on the Henry Ford Hospital Headache Disability Inventory (HDI, 100-point scale), de Roos identified reductions in disability score by 4.4 points in the probiotic group and by 3.7 points in the placebo group, which was not statistically different between the study groups.

## 4. Discussion

Many publications have examined the role of the gut (and the gut microbiome) in extra-intestinal diagnoses; indeed, the microbiome–gut–brain axis is one of the current hot topics in both clinical and pre-clinical research. Although our understanding of this important axis has progressed substantially in recent years, the role played by the gut microbiota in migraine pathophysiology remains to be convincingly established. This situation is reflected by the currently limited number of human clinical studies investigating the gut microbiota in migraine patients and how its manipulation might impact on its course.

Does the gut microbiome of migraineurs have unique features, which are reproducible? The exact nature the gut microbiome of the typical migraine patient is yet to be fully understood. Analysis of participants from The American Gut Project identified significant differences in nitrate, nitrite, and nitric oxide reductase genes (bacterial origin: genera Streptococcus and Pseudomonas) in the oral and fecal samples of migraineurs vs. non-migraineurs [29]. Nitrate compounds are well known headache triggers and they may cause migraines via release of substances such as calcitonin gene-related peptide (CGRP) [30,31]. Similarly, in the 2018 publication, Georgescu et al. demonstrated greater levels of dysbiosis in irritable bowel syndrome (IBS) patients with migraine when compared to IBS controls without a history of migraine [32]. While these two papers illustrate the need for further work to fully characterize the microbiome of migraineurs, they also provide the first evidence that differences exist in the microbiome of patients that experience migraine when compared to migraine-free controls. Neither of the randomized control trials (RCTs) included in this systematic review’s final analysis reported microbiome analysis of their participants—given the lack of definitive data around the gut microbiome of migraineurs, this would be a useful addition to any future probiotic research in migraines.

Interestingly, the findings from the two papers included in this systematic review were in sharp contrast to one another. The first of the two papers, by de Roos et al. [27], failed to demonstrate a statistically significant change in migraine frequency between the probiotic intervention group and placebo. In contrast, the study by Martami et al. [28], demonstrated an approximately 40% reduction in migraine frequency in the probiotic groups. In their study of EM patients, Martami et al. [28] saw a reduction in migraine frequency of 2.6 days per month compared to placebo. To put this into context, in a recent Cochrane review and meta-analysis of topiramate as a migraine prophylaxis for EM patients, Linde et al. [33] reported that the difference in effect between topiramate and placebo in the combined analysis corresponded to a ‘reduction in headache frequency of a little more than one headache per 28 days’.

The Martami study demonstrates a statistically and clinically significant reduction in migraine frequency in both EM and CM patients given the probiotic intervention, when compared to placebo. Both studies included in this systematic review sought to demonstrate the potential mechanism of action that might underpin any improvement in symptoms. Migraine is frequently thought of as a pro-inflammatory condition, while many probiotic species have widely acknowledged anti-inflammatory effects. As such, both sets of researchers measured serum inflammatory markers TNF-α and CRP, and the de Roos study also measured IL-6 and IL-10. Unfortunately, there were no clear trends evident in either data sets. Interestingly, the only statistically significant change in inflammatory markers was seen in the CM probiotic group in the Martami study. This particular population saw large reductions in migraine frequency, but counter-intuitively saw a statistically significant increase in CRP, although the clinical significance of this small absolute change (1 mg/dL) would be questionable, especially given that many institutions consider CRP results <10 mg/dL as within the normal range. It should also be noted that both CRP and TNF are known to vary based upon biological sex, race, and certain polymorphisms, and in addition to this, day-to-day variation in healthy populations has been observed at similar levels to that seen in this study [34,35,36]. Understanding the underlying mechanism of action for the improvements seen in the Martami study will be important for any future research into probiotics and migraine.

The multistrain probiotic products used in the de Roos and Martami studies differed in both colony forming unit (CFU) concentration and strain composition. Although some evidence of a dose response relationship exists within probiotic clinical research (e.g., antibiotic associated diarrhea), the current literature does not reveal a homogeneous picture [37]. Both probiotic consortia included a mix of *bifidobacteria* species and *lactobacilli*. If future studies seek to better characterize the typical microbiome of migraineurs, this could help to inform which of the microorganisms included here might be more—or less—effective at improving migraine symptoms. Microbiome and probiotic research is conducted at strain level, due to the strain-specific phenotypes and effects seen within probiotic species. Genomic and functional analysis of 100 strains of the species *L. rhamnosus*, for instance, clearly identified highly variable functional properties, such as ability to bind to mucus, resistance to bile salts, and adaptive immune capabilities [38]. Indeed, clinical research has shown that two strains of the same species may exert completely different effects on the host [38]. As such, more rigorous multi-omics analysis techniques may be necessary to inform this future strain selection.

Better understanding of the strain-specific functions of the chosen probiotic consortia will also help to understand the likely mechanism of action. Migraine pathophysiology is complex and remains to be definitively characterized. Multiple lines of evidence hint at the role of bacteria in migraine pathogenesis, however, which elements are more, or less, important remains to be defined. For example, multiple lines of evidence point towards an overlap between migraine and certain GI diagnoses, suggesting that an impaired gut barrier function may play a role in migraine pathogenesis [39]. Impaired gut barrier function can lead to an increase in the permeability of the gut, allowing the passage of bacteria and pro-inflammatory compounds such as lipopolysaccharides (LPS) into the systemic circulation; once in the systemic circulation, LPS is known to have a pro-inflammatory effect and can sensitize pain receptors in trigeminal sensory neurons [19]. Equally important is consideration of the role of serotonin in migraine pathophysiology—and the role of the gut microbiota in producing and modulating serotonin; both within the gut and within the CNS. Intra-ictally, migraine is known to be a low serotonin state and evidence suggests that depletion of tryptophan (the precursor to serotonin) exacerbates migraine symptoms [40]. Multiple probiotic strains produce serotonin or modulate the metabolism of tryptophan—all of which could be involved in the beneficial effects seen in the Martami study.

Given the inconsistent nature of the reporting of the two studies included in this systematic review, it would be helpful for future studies to include outcome measures that are easily comparable both with other microbiome interventions and also with conventional pharmacological migraine studies. Based on the studies conducted to date, the following set of recommendations, as shown in Table 3, could be adopted for future migraine studies.

## 5. Conclusions

With nearly 70 trials of probiotics in migraine, there is a clear lack of consistency in the reporting of results and as such, the results to date do not allow for effective meta-analysis. Therefore, standardization of methodology and reporting in future trials will be essential. To facilitate this, recommendations have been made in this systematic review to allow for meaningful comparisons between studies and between probiotic and conventional pharmacological interventions for migraine prophylaxis.

## Figures and Tables

**Figure 1 jcm-08-01441-f001:**
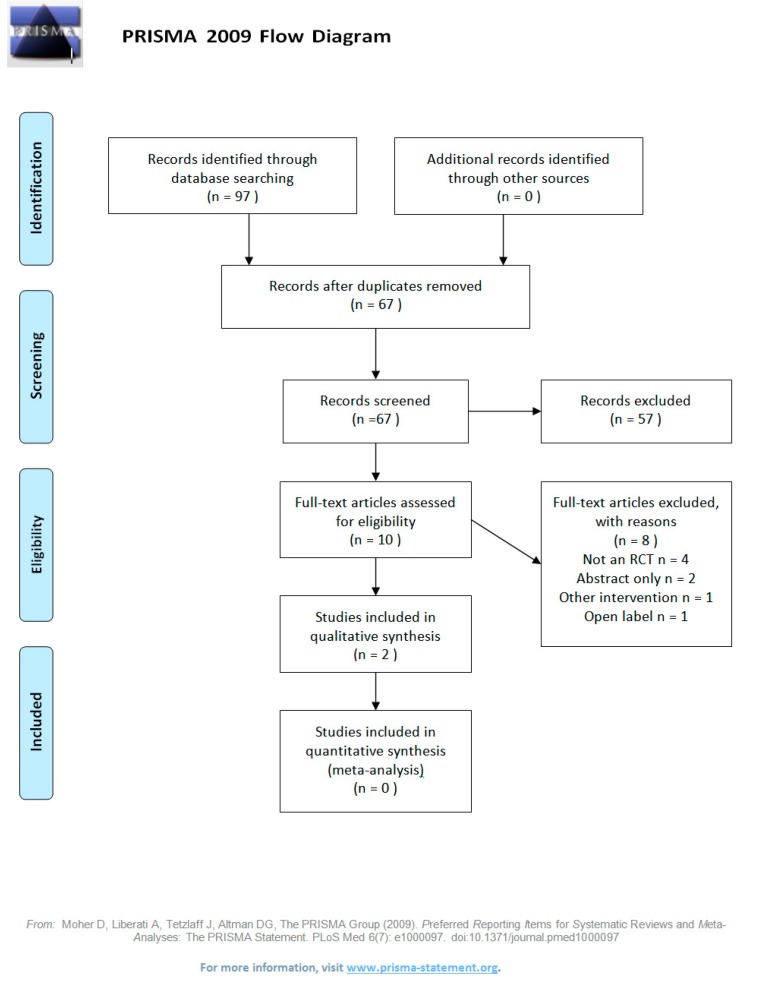
PRISMA 2009 flow diagram.

**Table 1 jcm-08-01441-t001:** Description of included studies.

	De Roos 2017 [27]	Martami 2019 Chronic [28]	Martami 2019 Episodic [28]
Design	Double blind RCT	Double blind RCT	Double blind RCT
Total completed n (F/M) Probiotic n (F/M) Placebo n (F/M)	60 (56/4) 31 (28/3) 29 (28/1)	40 (35/15) 23 (15/7) 18 (13/5)	39 (37/13) 21 (17/4) 18 (12/6)
Drop-out rate %	5% 3/63 (all placebo arm)	20% 10/50 (3 probiotic, 7 placebo)	22% 11/50 (4 probiotic, 7 placebo)
Follow-up	12 weeks	8 weeks	10 weeks
Migraine diagnostic criteria	ICHD-II	ICHD-III beta	ICHD-III beta
Migraine subtypes included	Episodic migraine	Chronic migraine	Episodic migraine
Probiotic	Ecologic Barrier, Winclove *Bifidobacterium bifidum* W23 *Bifidobacterium lactis* W52 *Lactobacillus acidophilus* W37 *Lactobacillus brevis* W63 *Lactobacillus casei* W56 *Lactobacillus salivarius* W24 *Lactococcus lactis* W19 *Lactococcus lactis* W58	Bio-Kult, Protexin *Bacillus subtilis* PXN 21 *Bifidobacterium bifidum* PXN 23 *Bifidobacterium breve* PXN 25 *Bifidobacterium infantis* PXN 27 *Bifidobacterium longum* PXN 30 *Lactobacillus acidophilus* PXN 35 *Lactobacillus delbrueckii ssp. bulgaricus* PXN 39 *Lactobacillus casei* PXN 37 *Lactobacillus plantarum* PXN 47 *Lactobacillus rhamnosus* PXN 54 *Lactobacillus helveticus* PXN 45 *Lactobacillus salivarius* PXN 57 *Lactococcus lactis ssp. lactis* PXN 63 *Streptococcus thermophilus* PXN 66	Bio-Kult, Protexin *Bacillus subtilis* PXN 21 *Bifidobacterium bifidum* PXN 23 *Bifidobacterium breve* PXN 25 *Bifidobacterium infantis* PXN 27 *Bifidobacterium longum* PXN 30 *Lactobacillus acidophilus* PXN 35 *Lactobacillus delbrueckii ssp. bulgaricus* PXN 39 *Lactobacillus casei* PXN 37 *Lactobacillus plantarum* PXN 47 *Lactobacillus rhamnosus* PXN 54 *Lactobacillus helveticus* PXN 45 *Lactobacillus salivarius* PXN 57 *Lactococcus lactis ssp. lactis* PXN 63 *Streptococcus thermophilus* PXN 66
Dose	2.5 × 10^9^ CFU/g, one sachet 2 g	2 × 10^9^ CFU/capsule, 2 capsules	2 × 10^9^ CFU/capsule, 2 capsules
Intake	Daily	Daily	Daily
Use of medications	Continued as usual	Continued as usual	Continued as usual
Outcomes measured	Migraine questionnaires and diaries (MIDAS, HDI); inflammation markers (IL, CRP, TNF); intestinal permeability (lactulose/mannitol test, zonulin levels in feces and serum)	Migraine questionnaires and diaries (migraine duration, severity, frequency, use of abortive medications); inflammation markers (CRP, TNF)	Migraine questionnaires and diaries (migraine duration, severity, frequency, use of abortive medications); inflammation markers (CRP, TNF)

F—female; M—male; ICHD—international classification of headache disorders; CFU—colony forming units; MIDAS—migraine disability assessment test; HDI—headache disability inventory; IL—interleukin; CRP—C-reactive protein; TNF—tumor necrosis factor; RCT—randomized control trial.

**Table 2 jcm-08-01441-t002:** The serum level of inflammatory markers at baseline and end of study.

	De Roos 2017 [27]		Martami 2019 Chronic [28]		Martami 2019 Episodic [28]	
	Probiotic	Placebo	Probiotic	Placebo	Probiotic	Placebo
TNF before (pg/mL)	2.45 ± 0.55	2.70 ± 1.2	5.90 ± 6.25	3.12 ± 3.46	2.97 ± 5.09	2.31 ± 3.51
TNF after (pg/mL)	2.57 ± 0.55	2.63 ± 0.67	8.18 ± 7.89	5.73 ± 5.38	2.73 ± 5.25	5.05 ± 6.94
CRP before (mg/dL)	2.0 ± 2.9	2.6 ± 3.3	1.77 ± 2.90	0.57 ± 0.73	1.09 ± 1.87	0.55 ± 0.4
CRP after (mg/dL)	1.8 ± 2.5	2.4 ± 3.1	2.70 ± 4.03 *	0.74 ± 1.01	0.83 ± 1.19	0.73 ± 0.33

Note: Data are presented as mean ± standard deviation. * Significantly different to result before intervention in the same group (*p*-value < 0.01). Paired *t*-test used to compare pre- and post-tests in all cohorts.

**Table 3 jcm-08-01441-t003:** Recommendations for future probiotic trials in migraine sufferers.

*Recommendations for future probiotic trials in migraine sufferers*
**Exclusion criteria**
Standardized exclusion criteria: use of medications with well-acknowledged impacts on the gut microbiome should be considered (e.g., antibiotics, antidepressants, proton pump inhibitors), as well as duration of the period free from antibiotics and probiotics.
Pregnancy and breastfeeding (due to the influence of hormonal changes).
**Inclusion criteria**
Studies should explicitly focus on chronic or episodic migraine sufferers and clearly indicate which group was studied.
Studies should focus on migraine with aura and without aura and clearly indicate which group was studied.
Gender factor should be considered. Many studies had only small proportion of male participants. Single gender studies could also be considered.
Standardized evaluation of the symptoms in the defined period prior to enrolment, minimum one month history. Ideally, to avoid reliance on patients’ memory, the first month of the trial would not include the intervention and allow for observation of the symptoms.
**Intervention**
Convenient and easy-to-use formulations should be chosen, to ensure good compliance.
Duration of intervention should be no less than two months.
Washout period should be studied to establish for each formulation how long after treatment cessation effects starts to diminish.
**General**
Use of validated standardized migraine symptom questionnaires/diaries for recording of migraine frequency (frequency of migraine onsets), duration (hours), and severity (validated scale or a comparable 10-point visual analogue scale (VAS)).
Validated Quality of Life tools.
If applicable, sub-group analysis comparing the effect of intervention in groups with different severities of migraine symptoms.
Inclusion of standard reporting metrics, such as 50% responder rates.
Standardized reporting on the use of medications pre and post intervention (number of doses per week, with the information what a standard local dose is).
Sub-group analysis for responders and non-responders, with explicit definition of responders (e.g., reduction of migraine duration by over 2 days per month).
Preferably, microbiome analysis pre- and post-intervention.
Consider inclusion of metabolomics outcomes such as lipopolysaccharides (LPS) and tryptophan.

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
