# Peer review of "Probiotics for the Prophylaxis of Migraine: A Systematic Review of Randomized Placebo Controlled Trials"

_jcm, 2019, doi:10.3390/jcm8091441_

Round 1

Reviewer 1 Report

In the manuscript titled “Probiotics for the prophylaxis of migraine: a systematic review of randomized placebo controlled trials”, Naghibi et al. provided a systematic review on the topic which is concise and structured; thus, very minor corrections are needed. Moreover, there are little to no grammatical errors in this article.

See below for general comments on the content:

Materials and Methods – Please provide the statistical analysis types to obtain the p-values which lead to the claim of the results being statistically significant. Table 2 – Please provide the measure of dispersion. Is it mean ± SD or mean ± SEM? Results – Despite the significant reduction of migraine days and severity in the CM group (Lines 182 and 185), the TNF and CRP markers were shown to be elevated when comparing the CM group to the placebo group in the Martami study – all of which were elaborated in the Discussion section. Despite mentioning that the statistically significant difference might be clinically insignificant, might the counter-intuitive increase in the inflammatory markers – especially TNF – be caused by sex differences (10.1016/j.jacc.2005.04.051), different polymorphisms, and receptor functionality (10.1177/0333102411419022)? Discussion – Given the different composition and dose of the probiotics being administered daily in the De Roos and the Martami studies, might this result in the significance seen in both Martami studies but not in the De Roos study?

See below for some specific comments on the formatting of the article:

Line 44 – Missing punctuation in “… during the reproductive years The pathophysiology …”. Line 123 – As RCT is an abbreviation of randomized control trial, the expansion should be included preceding the first use of the abbreviation. Figure 1 – Green lines appear to underline several words within the textboxes. The authors should correct this to improve the quality of the figure. Line 145 – A reference source seems to be missing.

Author Response

Replies to Reviewer 1 comments (replies in blue):

Comments and Suggestions for Authors

In the manuscript titled “Probiotics for the prophylaxis of migraine: a systematic review of randomized placebo controlled trials”, Naghibi et al. provided a systematic review on the topic which is concise and structured; thus, very minor corrections are needed. Moreover, there are little to no grammatical errors in this article.

See below for general comments on the content:

Materials and Methods – Please provide the statistical analysis types to obtain the p-values which lead to the claim of the results being statistically significant.

When referring to statistical significance, this was in each case referring back to the original papers, not to any analysis or meta-analysis conducted for this Sytematic Review. In light of this we have not made any changes to the manuscript. 

Table 2 – Please provide the measure of dispersion. Is it mean ± SD or mean ± SEM?

Data are presented as mean +/- standard deviation.

*Paired t-test was used to compare pre-post tests.

These changes have been added to the manuscript

Results – Despite the significant reduction of migraine days and severity in the CM group (Lines 182 and 185), the TNF and CRP markers were shown to be elevated when comparing the CM group to the placebo group in the Martami study – all of which were elaborated in the Discussion section. Despite mentioning that the statistically significant difference might be clinically insignificant, might the counter-intuitive increase in the inflammatory markers – especially TNF – be caused by sex differences (10.1016/j.jacc.2005.04.051), different polymorphisms, and receptor functionality (10.1177/0333102411419022)?

We have added these two points to the discussion section and also included the above references.

Discussion – Given the different composition and dose of the probiotics being administered daily in the De Roos and the Martami studies, might this result in the significance seen in both Martami studies but not in the De Roos study? 

Again, we have amended the discussion to draw out these points of dose-dependent response and strain-specific responses. 

See below for some specific comments on the formatting of the article:

Line 44 – Missing punctuation in “… during the reproductive years The pathophysiology …”. Thank you for spotting this - it has been changed.

Line 123 – As RCT is an abbreviation of randomized control trial, the expansion should be included preceding the first use of the abbreviation. Again, thank you - now changed.

Figure 1 – Green lines appear to underline several words within the textboxes. The authors should correct this to improve the quality of the figure. Figure amended

Line 145 – A reference source seems to be missing. Apologies but I am unsure what this refers to. I have checked repeatedly but am still unsure.

Reviewer 2 Report

The manuscript is interesting in showing the doubt on probiotics in migraine. The method of evaluating the studies in correct.

I may suggest to add after ref 24 the following ref on the current advances in therapy of migraine to give an idea of the status of art: 

Recent advances in migraine therapy.

Antonaci F, Ghiotto N, Wu S, Pucci E, Costa A.

Springerplus. 2016 May 17;5:637. doi: 10.1186/s40064-016-2211-8. eCollection 2016. Review.

Author Response

Replies to Reviewer 2 comments (replies in blue):

Comments and Suggestions for Authors

The manuscript is interesting in showing the doubt on probiotics in migraine. The method of evaluating the studies in correct.

I may suggest to add after ref 24 the following ref on the current advances in therapy of migraine to give an idea of the status of art: 

Recent advances in migraine therapy.

Antonaci F, Ghiotto N, Wu S, Pucci E, Costa A.

Springerplus. 2016 May 17;5:637. doi: 10.1186/s40064-016-2211-8. eCollection 2016. Review.

Thank you, this is a really interesting paper and we have added it as reference 25.